# Biscuits Polyphenol Content Fortification through Herbs and Grape Seed Flour Addition

Ondřej Král, Zdeňka Javůrková [ID], Dani Dordevic [ID], Matej Pospiech *[ID], Simona Jančíková [ID], Kseniia Fursova and Bohuslava Tremlová [ID]

Department of Plant Origin Food Sciences, Faculty of Veterinary Hygiene and Ecology, University of Veterinary Sciences Brno, Palackeho Tr. 1946/1, 612 42 Brno, Czech Republic; H18006@vfu.cz (O.K.); javurkovaz@vfu.cz (Z.J.); dordevicd@vfu.cz (D.D.); jancikovas@vfu.cz (S.J.); fursova-k@mail.ru (K.F.); tremlovab@vfu.cz (B.T.)
* Correspondence: mpospiech@vfu.cz

**Abstract:** The study aimed to verify whether the addition of selected herbs and spices will affect the content of polyphenols in biscuits and their antioxidant capacity, as well as what impact it will have on their sensory properties and attractiveness to consumers. Ground cloves, cinnamon, mint, and grape flour were added to the biscuits in concentrations of 1.0, 3.0, 5.0, and 10.0%. The total content of polyphenols in spices and biscuit samples was determined using the Folin–Ciocalteau solution and, subsequently, the antioxidant capacity was measured by FRAP (ferric ion reducing antioxidant power) and DPPH (2,2-diphenyl-1-picrylhydrazyl inhibition). Polyphenols were transferred through spices and herbs into the biscuits in all samples and thus their antioxidant capacity was increased. The antioxidant capacity of the control sample measured by the DPPH method was 15.41%, and by the FRAP method 1.02 μmol Trolox/g. There was an increase in antioxidant capacity in all samples with the addition of spices and herbs. The highest increase was recorded in the sample with cloves, namely with the addition of 10% of cloves there was an increase measured by the DPPH method to 92.6% and by the FRAP method to 208.42 μmol Trolox/g. This also corresponds to the measured TPC (Total Polyphenol Content) in the pure clove, which was 219.09 mg GAE/g, and in the samples where the content gradually grew up to 4.51 mg GAE/g in the sample with the addition of 10%, while the polyphenol content of the control sample was 0.2 mg GAE/g. For other parameters, changes were also observed, depending on the addition of spices/herbs. There was a reduction in both texture parameters, hardness and fracturability, depending on the addition of spices/herbs, which was confirmed by both instrumental measurements and sensory analysis. Colour measurements clearly separated the control from the fortified samples, thus confirming the colour changes. The addition of grape flour shows the smallest difference from the control when the overall impression does not change with the addition. In terms of the combination of increased antioxidant capacity and overall consumer acceptability, the addition of cloves at a concentration of 3.0% appears to be the best option.

**Keywords:** biscuits; texture; polyphenols; spices; herbs; sensory; antioxidants



## 1. Introduction

Biscuits are popular for their sensory profile and ease of consumption, but also for their ease of storage and long shelf life. As biscuits are one of the most commonly consumed sweets, by adjusting their nutritional profile, it is possible to influence the diet of a rather wide human population [1]. However, due to their higher fat content and ambient storage conditions, they undergo changes that affect their quality. One of them is oxidation, in which hydroxyperoxides are first formed, causing a number of secondary reactions with evolution of aldehydes, ketones, acids, and other low-molecular-weight volatile substances. The ensuing decrease in the flavour and colour quality, which renders a food unappetizing in appearance, and the accumulation of toxic reaction products pose a major problem to the food industry [2]. Synthetic antioxidants, such as butylated hydroxytoluene (BHT) and

butylated hydroxyanisole (BHA), were used frequently to slow this oxidation, but their use is limited due to safety concerns [3]. Thus, there is a tendency these days to use natural antioxidants. Numerous herbs and spices have the potential to slow the oxidation of lipids during food storage. Recently, the use of natural antioxidants in the food industry has expanded rapidly, and many related studies have been published [4]. A study of the effect of a mint extract addition on the oxidation and overall sensory properties of biscuits was performed. The extract was also compared with the BHA, which demonstrated the antioxidant effect of mint [5]. The antioxidant activity of herbs and spices is mainly due to the content of phenolic diterpenes, flavonoids, volatile oils, and phenylpropanoids. Phenolic compounds are bioactive substances widely present in plants and spices. These compounds are of interest to their multiple biological activities such as free radical scavenging capacity, metal chelation, inhibition of cellular proliferation, modulation of signal transduction pathways, and enzymatic activity [6,7]. These compounds are therefore considered suitable candidates to prevent lipid oxidation. The presence of an antioxidant is one of the fastest ways to reduce fat oxidation [8]. In the case of biscuits, it is, therefore, appropriate to choose plants and spices with a high content of phenolic substances and at the same time not negatively affecting the sensory parameters and the attractiveness of the product for the customer. There are a number of studies examining the effects of these natural substances in various foods [7,9].

Food perception is a multisensory experience that combines information gained through sight, taste, smell, hearing, and touch. The term "visual appearance of food" consists of many attributes such as colour, shape, texture, gloss, size, and variety. The perceptions of food perceived in this way determine the attractiveness of the product, sensory quality, aesthetics, the expected safety, willingness to accept the product, and the attractiveness of the taste, and all of these factors influence the consumer's choice [10]. The colour of biscuits is one of the important factors influencing the acceptability of biscuits for consumers. Although the assessment of colour, as well as other organoleptic parameters, is not objective, it is often crucial for a large proportion of consumers. Colour can be determined by simple comparison between samples or using instrumental methods such as colourimetry and spectrophotometry [11,12]. Image analysis methods are flexible and can be a possible substitute for the decision-making process based on human vision [13,14]. Although the processing of image information by the human eye has several advantages, it has a major disadvantage in the subjectivity of human observation for objective evaluation (size, shape, colour of objects, etc.) [15]. Currently, image analysis is a widely used technique; it also has its place in the field of monitoring defects in cereal products during production or storage processes [16].

Based on previous published studies, we chose the following natural materials, for which we expected a positive effect on the overall quality of biscuits. These were cinnamon powder (*Cinnamomum verum*), crushed mint leaves (*Mentha piperita* L), ground cloves (*Eugenia caryophyllata*), and grape seed flour (*Vitis vinifera*). This study aims to verify whether these natural materials can affect the antioxidant activity in biscuits and to find the optimal addition that is still sensorily acceptable to consumers.

## 2. Materials and Methods

### 2.1. Preparation of Biscuits

The biscuits were prepared according to commercial formula (Penam a.s., Brno, Czech Republic). The following ingredients were used: wheat plain flour (150 g; Předměřická mouka hladká, J. Voženílek, Předměřice nad Labem, Czech Republic), baking fat (45.0 g; Hera Classic, Upfield, Prague, Czech Republic), powdered sugar (50.0 g; Cukr moučka, TTD, Dobrovice, Czech Republic), and egg yolk (18.0 g). Wheat plain flour, which is preferred [17] in the production of biscuits, was chosen for the test. The ingredients were mixed for 3 min in a food processor KVL8320S (Kenwood, Havant, UK) and the mass was then shaped into round biscuits (10.0 g/piece) and baked in a hot air oven ELENA (Unox, Cadoneghe, Italy) for 19 min at 180 °C. The control sample was marked as SC. Other

samples were prepared in the same manner, and the following ingredients were added one by one: ground cloves (*Eugenia caryophyllata*, Vitana, Czech Republic, sample marked as SH), ground cinnamon (*Cinnamomum verum*, Vitana, Czech Republic, sample marked as SS), mint stalks (*Mentha piperita* L., Vitana, Czech Republic, sample marked as SM), and grape seed flour (*Vitis vinifera*, Zdraví z přírody, Czech Republic, sample marked as SHo) in concentrations of 1.0%, 3.0%, 5.0%, and 10.0%.

### 2.2. Reagents Used

The reagents for total polyphenols determination were $Na_2CO_3$; Folin–Ciocalteau; Gallic acid 100% p.a. quality (PENTA, Brno, Czech Republic) The following reagents and purity were used to determine the antioxidant capacity by the FRAP method: acetate buffer—300 mM/L 2,4,6-tris-2-pyridyl-1,3,5-triazine, 99% (VWR, Stříbrná Skalice, Czech Republic); HCl 35%, p.a. (PENTA, Brno, Czech Republic); FeCl3 6H2O, pur. (Lachema, Brno, Czech Republic); Methanol 99.5%, p.a. (Lach-ner, Neratovice, Czech Republic); and Trolox 99% (VWR, Stříbrná Skalice, Czech Republic). For DPPH method, the following reagents were used: Ethanol 96%, p.a. (Threos, Kojetín, Czech Republic); 2,2-Diphenyl-1-picrylhydrazyl (DPPH); 95% (VWR, Stříbrná Skalice, Czech Republic).

### 2.3. Assessment of Total Polyphenols Content

The determination of total polyphenols content was made using the Folin–Ciocalteau solution by Sarker and Oba [18], with some modifications. Briefly, the samples were homogenized and then 10.0 mL of distilled water was added in the beaker with 1.0 g of the sample. After 10 min of shaking at 150 rpm, Li-3 (Nedform s.r.o., Valašské Meziříčí, Czech Republic), the sample was filtered (Whatman grade 1). 1.0 mL of the filtered extract was pipetted for the reaction and 5 mL of Folin–Ciocalteau solution and 4.0 mL of $Na_2CO_3$ (75.0 g/L) were added. After 30 min of incubation in the 25 mL volumetric flask in the dark, the absorbance was measured at 765 nm by the spectrophotometer (CE7210 Diet-Quest, Cambridge, UK). The solution with 1.0 mL of water instead of the sample was used as the blank. Gallic acid was used as the standard. Assessment in detail in Supplementary Materials Protocol S1.

### 2.4. Assessment of Antioxidant Capacity

The antioxidant capacity was measured by the DPPH (2,2-diphenyl-1-picryl-hydrazyl) and FRAP (Ferric reducing ability of plasma) analysis.

Using the FRAP method, 0.1 g of a homogenized sample of biscuits (0.01 g in the case of pure spices) in 20.0 mL of 75% methanol was extracted by ultrasound DT 510H (Bandelin, Berlin, Germany). Then, 180.0 μL of the filtered sample was removed, 300.0 μL of distilled water, and 3.6 mL of working solution (acetate buffer + TPTZ + $FeCl_3 \times 6H_2O$ in ratio 10:1:1) were added, and this mixture was incubated for 8 min in the dark. Subsequently, the absorbance at 593 nm was measured by the spectrophotometer CE7210 (Diet-Quest, Cambridge, UK). The results are expressed in ug/mL of Trolox, which was used for the calibration curve [19].

For the DPPH method [20], the samples were homogenized, and 0.1 g was weighted in dark glasses (0.01 g in the case of pure spice). Next, 20 mL of 96% $v/v$ ethanol was added, and the sample was in ultrasound for 30 min. Then, the 3.0 mL of extracts and 1.0 mL of 0.1 mM ethanolic solution of DPPH was mixed in the test tube and incubated in dark for 30 min. The absorption was measured at 517 nm against ethanol by the spectrophotometer CE7210 (Diet-Quest, Cambridge, UK). The radical scavenging activity was calculated by the equation Equation (1).

$$\text{DPPH scavenging activity [\%]} = ((\text{AbsDPPH} - \text{Abssample})/\text{AbsDPPH}) \times 100 \quad (1)$$

### 2.5. Texture Assessment

Texture determination involved hardness and friability measurements. The TA.XT.PLUS Texture Analyzer (Stable Micro Systems, Godalming, UK) was used for the measurement. The whole biscuit was placed in a Circular Support Rig (A/CS) and a 5.0 mm penetration probe (SMS P/5) was used to penetrate the sample in the centre area similar to. The probe penetrated the samples 2.0 mm deep, the penetration rate before the test was set to 1.5 mm/s, while the speed after the test was 0.50 mm/s.

### 2.6. Sensory Analysis

The sensory properties of the products were assessed by a panel of 15 trained and tested evaluators. These assessors were individuals between the ages of 20 and 55 selected from university staff and students who had passed sensory tests. The samples were identified by a three-digit code. The evaluation took place in a laboratory equipped in compliance with the ISO 8589 standard [21]. The parameters of colour, consistency, smell, taste, sweetness, overall impression, and the price that the evaluators would be willing to pay for the product were evaluated. All evaluations were recorded by means of a questionnaire survey on a scale from 0 to 100, where higher value means a higher intensity.

### 2.7. Colour Measurement

The colour analysis of biscuits was completed with a Canon EOS 600D camera (Canon, Tokyo, Japan). For the standard light conditions, two DELUX L—1 × 18 W (Osram, München, Germany) lamps were used. For standard position, the Fomei CS 920 tripod was used. The capturing conditions were as follows: exposure time 1/40; F-Number—5.6; exposure program—Manual; ISO speed ratings—200; Aperture—F 5.7; format RAW; resolution 5184 × 3456; 24 bit sRGB. Each biscuit was captured 3 times and the images were analysed by the image analysis software of NIS-Elements BR 5.20.00. RGB colour channels and recalculated HSV were measured. The used features included Mean Intensity, Intensity Variation, Mean Red, Mean Green, Mean Blue, Hue Typical, Hue Variation, Mean Saturation, Mean Brightness, Brightness Variation, Mean Density, and Density Variation.

### 2.8. Statistical Analysis

For the comparison of fortified biscuits, the data was evaluated by one-way ANOVA, and the Multiple Comparisons with t Distribution post hoc nonparametric test was used to find differences between individual groups. The IBM statistics computer program SPSS 2010 (IBM Corp, Armonk, NY, USA) was used. The sensory analysis was evaluated by the non-parametric Kruskal–Wallis test. Statistical significance of the obtained results was determined at the significance level of $\alpha = 0.05$. For the colour comparison, PCA and loadings plot analysis of XLSTAT 2021 (Addinsoft, Paris, France) was applied.

## 3. Results and Discussion

### 3.1. Determination of Antioxidant Capacity and Total Amount of Polyphenols

Maintaining the high quality of biscuits is of great economic importance, as they are often stored for a rather long period of time. The onset of rancidity in baked goods affects the texture, colour, and other sensory properties. If consumers' expectations regarding texture, colour, and sensory parameters are not met, there will be no interest in the product and it will not be sold [9,22]. The herbs and native spices used have a naturally high content of polyphenols and thus a high antioxidant capacity. They can, therefore, be used to slow down oxidation processes. Baked products have a significant reduction in the content of polyphenols [23] and can therefore be suitably supplemented by the addition of spices. Another study [9] examined the effect of mint addition on sensory properties, texture, and colour, in the form of powder, pure menthol, and mint extract. It was found that the addition of mint powder was the most suitable. The effect of adding raisin extract instead of synthetic antioxidants to prolong the shelf life of biscuits was also tested. The results show that the addition of 1.0 to 2.0% does not affect the organoleptic properties

According to our measurements, the addition of appropriately selected spices and herbs affects the oxidation of biscuits.

The results obtained for the content of polyphenols, DPPH and FRAP clearly show a statistically significant dependence ($p < 0.05$) between samples prepared with different additions of selected herbs and spices. Their addition demonstrably ($p < 0.05$) increases the content of polyphenols in biscuits, and their increase depends on the concentration of these herbs and spices in the products. The results are presented in Table 1. Clove has the highest content of polyphenols, as shown both in the spice itself and in biscuit samples. The measured value of 219.09 mg GAE/g in the clove also corresponds to the values measured in other studies [7,23]. The antioxidant effect of clove oil was also tested in a baked cake. This oil was compared with a synthetic antioxidant for 28 days and demonstrated high antioxidant activity and effective antimicrobial properties in tests [4]. In this study, the antioxidant capacity of a sample containing cloves determined by the DPPH method increases up to the addition of 3.0%, and further changes are no longer statistically significant ($p > 0.05$). In the case of FRAP determination, the antioxidant capacity of samples with cloves increases, even for concentrations of 5.0% and 10.0%, and is the highest of all the examined samples. The measured content of polyphenols 101.59 mg GAE/g for cinnamon is significantly higher than reported in the study by Ali, Wu, Ponnampalam, Cottrell, Dunshea, and Suleria [7]. Differences in the total polyphenol content may be due to the extraction methods used [24]. Cultivar differences and the geographical location of the spices where they were grown can also affect results [7,25]. The antioxidant capacity in these samples increased with both methods used, but the content of polyphenols at the concentrations of SS 1, SS 3, and SS 5 does not increase in proportion to the amount of spice addition. This discrepancy can be explained by the content of other natural flavonoids and essential oils in cinnamon [26,27]. Mint and grape seed flour have a significantly lower content of polyphenols. The measured polyphenol content of grape flour (5.75 mg GAE/g) corresponds to a previous study measuring 5.58 mg GAE/g [28]. This low content of polyphenols is also reflected in the antioxidant capacity of biscuits prepared with the addition of this flour, which is the lowest of the examined samples. In the case of mint, the measured polyphenol content is 25.02 mg GAE/g. In other studies, a higher content was measured—109.98 mg GAE/g [29]—but the value measured by us corresponds typically to the measured values for different varieties 14.66–43.21 mg GAE/g [30]. The antioxidant capacity measured by the DPPH method at 20.57% also corresponds to the values measured in this study (7.5–44.66%). The difference in the content of polyphenols, and thus in the antioxidant capacity, is caused by the large variability of mint given by the species, the geographical location where the mint is grown, and the degree of leaves maturity. This, in turn, affects the chemical composition of the essential oils present, the bioactivity and the antioxidant capacity [5,31,32]. In the case of biscuits containing mint, the antioxidant capacity determined by the DPPH method increases with the amount of its addition, but when using the FRAP method, the maximum is reached already at a concentration of 3% and decreases with higher addition. The SHo effect to oxidative stability was confirmed by other authors as well [33]; the authors also confirmed a decreasing peroxide value as an effect of decomposition of hydroperoxides. Likewise, increase in the DPPH and FRAP was reported by other authors for the addition of cloves [34], cinnamon [35], and mint [36,37].

**Table 1.** Polyphenol content and antioxidant capacity measured by the DPPH and FRAP methods of experimentally produced biscuits.

| Samples | Polyphenols (mg GAE/g) | DPPH (%) | FRAP (µmol Trolox/g) |
|---|---|---|---|
| Biscuits with the addition of grape seed flour (*Vitis vinifera*) | | | |
| SC | 0.20 ± 0.00 [a] | 15.41 ± 1.16 [a] | 1.02 ± 0.01 [a] |
| SHo1 | 0.43 ± 0.00 [b] | 34.97 ± 1.02 [b] | 5.24 ± 0.05 [b] |
| SHo3 | 0.59 ± 0.00 [c] | 42.39 ± 2.35 [c] | 8.94 ± 0.08 [c] |
| SHo5 | 0.70 ± 0.00 [d] | 55.05 ± 5.96 [d] | 11.02 ± 0.11 [d] |
| SHo10 | 0.97 ± 0.00 [e] | 64.11 ± 3.60 [e] | 17.49 ± 0.13 [e] |
| Biscuits with the addition of peppermint (*Mentha piperita* L.) | | | |
| SC | 0.20 ± 0.00 [a] | 15.4 ± 0.001 [a] | 1.02 ± 0.01 [a] |
| SM1 | 0.62 ± 0.00 [b] | 44.8 ± 0.001 [b] | 6.70 ± 0.05 [b] |
| SM3 | 1.08 ± 0.00 [c] | 61.5 ± 0.000 [c] | 10.73 ± 0.08 [c] |
| SM5 | 1.49 ± 0.00 [d] | 67.0 ± 0.004 [c] | 9.11 ± 0.08 [d] |
| SM10 | 2.34 ± 0.00 [e] | 84.2 ± 0.000 [d] | 9.59 ± 0.07 [e] |
| Biscuits with the addition of cloves (*Eugenia caryophyllata*) | | | |
| SC | 0.20 ± 0.00 [a] | 15.4 ± 0.001 [a] | 1.02 ± 0.01 [a] |
| SH1 | 0.88 ± 0.00 [b] | 77.4 ± 0.003 [b] | 24.45 ± 0.08 [b] |
| SH3 | 1.60 ± 0.01 [c] | 90.5 ± 0.000 [c] | 25.08 ± 0.05 [c] |
| SH5 | 2.41 ± 0.05 [d] | 90.7 ± 0.001 [c] | 102.23 ± 0.65 [d] |
| SH10 | 4.51 ± 0.02 [e] | 92.6 ± 0.000 [c] | 208.42 ± 1.11 [e] |
| Biscuits with the addition of cinnamon (*Cinnamomum verum*) | | | |
| SC | 0.20 ± 0.00 [a] | 15.4 ± 0.001 [a] | 1.02 ± 0.01 [a] |
| SS1 | 0.41 ± 0.00 [b] | 40.9 ± 0.003 [b] | 2.54 ± 0.01 [b] |
| SS3 | 0.35 ± 0.00 [c] | 79.7 ± 0.002 [c] | 3.89 ± 0.04 [c] |
| SS5 | 0.42 ± 0.00 [d] | 81.5 ± 0.005 [c] | 10.77 ± 0.01 [d] |
| SS10 | 1.10 ± 0.00 [e] | 90.1 ± 0.000 [d] | 36.61 ± 0.30 [e] |
| Pure herbs and spices | | | |
| Peppermint (*Mentha piperita* L.) | 25.02 ± 0.01 | 20.57 ± 0.000 | 240.00 ± 0.38 |
| Cinnamon (*Cinnamomum verum*) | 101.59 ± 0.04 | 71.48 ± 0.001 | 1322.57 ± 18.08 |
| Cloves (*Eugenia caryophyllata caryophyllata*) | 219.09 ± 0.01 | 85.18 ± 0.002 | 3316.52 ± 14.22 |
| Grape seed flour | 5.75 ± 0.02 | 65.24 ± 0.61 | 225.23 ± 1.89 |

Different letters in columns indicate statistically significant ($p < 0.05$) differences; SC, control biscuit; SHo, biscuit with added grape seed flour (0, 3, 5, or 10%); SM, biscuit with added mint (0, 3, 5, or 10%); SH, biscuit with added clove (0, 3, 5, or 10%); or SS, biscuit with added cinnamon (0, 3, 5, or 10%) $p < 0.05$.

*3.2. Texture*

The measured values for hardness and fracturability are given in Table 2. The results show a statistically significant ($p < 0.05$) reduction in both parameters in the case of all additions of spices and herbs. These parameters are reduced already by the addition of 1.0%, and other changes are not statistically significant ($p > 0.05$). Only in the case of mint, with the addition of 10.0%, is there an increase again. This can be caused by larger parts of the leaves, which are then more concentrated at the penetration of the probe.

In all cases, the addition of spices and herbs reduced the hardness of the biscuits and their friability statistically significantly ($p < 0.05$). Except for mint, the parameters were changed at the lowest addition already, but did not change further with higher amounts. For mint, with the addition of 10.0%, the difference between the sample and the control for hardness and friability was statistically insignificant ($p > 0.05$). The reason is the presence of small leaf particles, which, at high concentrations in the biscuits, affect the results obtained by the method used. The decrease in hardness after the addition of SHo was also confirmed [33]. On the other hand, the decrease in textural parameters for SS was not in compliance with [38], where a similar parameter was slightly increasing. The significant differences between the control sample and samples with clove addition were also confirmed by penetration properties, such as hardness descriptors [34]. However, for the mint addition, the differences were not confirmed [37]. The assessment of changes

in chemical composition (water content, protein, sugar, fibre, etc.) due to the addition of spices was not the subject of this study. However, as reported by Bajaj, Urooj, and Prabhasankar [22], the addition of mint powder results in an increased content of fibre, protein, and ash, but a decreased content of moisture and sucrose, as well as a lower sensory evaluation. The results of this sensory evaluation show that the colour, crunchiness, and taste were acceptable until the addition of 4.0% cinnamon powder.

**Table 2.** Textural parameters in the samples of biscuits.

| Sample | Hardness (g) | Fracturability (g) |
|---|---|---|
| Biscuits with the addition of grape seed flour (*Vitis vinifera*) | | |
| SC | 10,068.8 ± 1098.25 [a] | 7737.46 ± 809.73 [a] |
| SHo1 | 2085.52 ± 737.36 [b] | 3924.94 ± 220.71 [b] |
| SHo3 | 1806.74 ± 446.09 [b] | 3798.7 ± 425.94 [b] |
| SHo5 | 2934.78 ± 584.75 [b] | 2689.79 ± 399.6 [b] |
| SHo10 | 2142.81 ± 197.44 [b] | 3458.87 ± 980.54 [b] |
| Biscuits with the addition of peppermint (*Mentha piperita* L.) | | |
| SC | 10,068.8 ± 1098.25 [a] | 7737.46 ± 809.73 [a] |
| SM1 | 3112.33 ± 1295.61 [b] | 4524.03 ± 288.96 [b] |
| SM3 | 4637.05 ± 880.37 [b] | 4468.19 ± 1382.39 [b] |
| SM5 | 3322.03 ± 1506.43 [b] | 4160.94 ± 882.68 [b] |
| SM10 | 9504.44 ± 1460.19 [a] | 7803.87 ± 828.18 [a] |
| Biscuits with the addition of cloves (*Eugenia caryophyllata*) | | |
| SC | 10,068.8 ± 1098.25 [a] | 7737.46 ± 809.73 [a] |
| SH1 | 3401.11 ± 1454.35 [b] | 3941.56 ± 667.56 [b] |
| SH3 | 1195.27 ± 550.54 [b] | 1794.38 ± 691.11 [b] |
| SH5 | 2153.17 ± 462.01 [b] | 2555.33 ± 516.5 [b] |
| SH10 | 2279.71 ± 1339.79 [b] | 2569.36 ± 1158.18 [b] |
| Biscuits with the addition of cinnamon (*Cinnamomum verum*) | | |
| SC | 10,068.8 ± 1098.25 [a] | 7737.46 ± 809.73 [a] |
| SS1 | 4917.92 ± 129.69 [b] | 4880.96 ± 992.69 [b] |
| SS3 | 2014 ± 412.89 [b] | 4312.43 ± 872.71 [b] |
| SS5 | 3004.65 ± 1139.35 [b] | 4179.81 ± 406.29 [b] |
| SS10 | 4045.75 ± 1738.87 [b] | 5747.69 ± 352.3 [b] |

Different letter in columns indicate statistically significant (*p* < 0.05) differences; SC, control biscuit; SHo, biscuit with added grape seed flour (0, 3, 5, or 10%); SM, biscuit with added mint (0, 3, 5, or 10%); SH, biscuit with added clove (0, 3, 5, or 10%); and SS, biscuit with added cinnamon (0, 3, 5, or 10%).

### 3.3. Sensory Analysis

The results of the sensory analysis are expressed in Table 3. The results show that, in the case of the addition of grape seed flour, no changes in taste, sweetness, overall impression, and price were found. Statistically significant (*p* < 0.05) differences were found in colour, consistency, and odour. The colour changes depending on the addition of spices/herbs are clearly shown in Figure 1. The colour and aroma were more intense, while the consistency was significantly (*p* < 0.05) lower than in the control sample without the addition of grape seed flour. The effect of grape seed flour primarily on the sensory properties of biscuits was reported by Ng and Wan Rosli [38]. In their study, it was demonstrated that the addition of more than 13.0% grape seed flour improved the fracturability and friability of biscuits. The water absorption of this flour at 30 °C is three times higher than the absorption of wheat flour. This also contributes to a better structure of the biscuits. The effect of grape seed flour has also been studied in other products, such as sausages. The addition of grape seed flour to these meat products led to a decrease in oxidation, probably due to the content of antioxidants in this flour. However, its addition significantly affected all sensory parameters. When more than 0.5% was used, the product was generally less acceptable [39].

No significant effects in taste and sweetness were observed in biscuits containing mint (*p* > 0.05). A significantly higher intensity of colour and odour was again observed

($p < 0.05$), while the consistency value decreased ($p < 0.05$). The colour increased depending on the addition, while the smell did not change with increasing the addition. In the case of the overall impression and price, there was a significant ($p < 0.05$) decrease compared to the control sample with the addition of 10% mint. This shows that the 10.0% addition is no longer attractive to consumers. Lower concentration up to 5% did not show any significant differences from the control ($p < 0.05$). This finding was also confirmed for 1.2% addition of mint into the buckwheat-based cookies [36]. However, the attractiveness can vary depending on consumers' or national preferences. Certain studies [22,37] report that already a 1% mint addition was evaluated as less attractive than the control and other sensory criteria were also involved.

Sweetness, overall impression, and price show a statistically significant ($p < 0.05$) reduction in SH samples with 10% addition of cloves. The decreasing of overall impression was also confirmed from the addition of 0.8% [34], whileour study confirmed reduction from the addition of 5% in comparison with the control. The taste does not differ statistically significantly ($p > 0.05$), but the high SD was recorded, and other study also confirms changes in the taste with clove addition [34]. In summary, the results show that the sample with the clove addition of 10.0% is no longer accepted positively.

Samples with the addition of cinnamon show a significant improvement in parameters ($p < 0.05$) of colour, aroma, taste, sweetness, overall impression, and price. However, with the addition of 10.0%, there is a significant ($p < 0.05$) decrease in the overall impression and price. Here, too, it is shown that this addition already reduces the attractiveness of the product. The consistency of the sample with the addition decreases ($p < 0.05$), but does not change further ($p > 0.05$), depending on the concentration of cinnamon.

**Table 3.** The sensory evaluation of biscuits.

| Sample | Colour | Consistency | Smell | Taste | Sweetness | Overall Impression | Price (Kč) |
|---|---|---|---|---|---|---|---|
| | | | | Biscuits with the addition of grape seed flour (*Vitis vinifera*) | | | |
| SC | 17.57 ± 12.66 [a] | 72.14 ± 16.89 [a] | 29.14 ± 13.86 [a] | 37.00 ± 17.57 | 35.43 ± 14.52 | 39.57 ± 22.26 | 7.86 ± 11.07 |
| SHo1 | 57.29 ± 9.72 [c] | 47.71 ± 7.93 [c] | 40.00 ± 11.37 | 29.14 ± 15.03 | 40.43 ± 13.83 | 34.57 ± 17.06 | 9.14 ± 10.21 |
| SHo3 | 68.14 ± 9.42 [b,c] | 45.29 ± 2.93 [b,c] | 42.14 ± 9.92 | 39.86 ± 13.75 | 43.29 ± 13.76 | 55.43 ± 14.12 | 18.71 ± 10.00 |
| SHo5 | 76.00 ± 11.17 [b] | 44.86 ± 7.43 [b,c] | 44.14 ± 16.36 [b] | 38.00 ± 19.58 | 42.43 ± 14.00 | 39.29 ± 22.48 | 11.00 ± 10.66 |
| SHo10 | 73.57 ± 13.81 [b] | 48.43 ± 6.88 [b] | 47.57 ± 10.77 [b] | 24.00 ± 6.98 | 36.14 ± 14.40 | 32.71 ± 19.66 | 6.43 ± 7.48 |
| | | | | Biscuits with the addition of peppermint (*Mentha piperita* L.) | | | |
| SC | 17.57 ± 12.66 [a] | 72.14 ± 16.89 [a] | 29.14 ± 13.86 [a] | 37 ± 17.57 | 35.43 ± 14.52 | 39.57 ± 22.26 [a] | 7.86 ± 11.07 [a] |
| SM1 | 49.5 ± 13.37 [c] | 45.67 ± 4.33 [b] | 54.92 ± 13.87 [b] | 44.17 ± 20.1 | 45.92 ± 11.59 | 48.25 ± 14.58 [a] | 21.42 ± 13.45 [b,c] |
| SM3 | 56.08 ± 11.58 [b,c] | 50.67 ± 6.8 [b] | 55.67 ± 13.87 [b] | 40.25 ± 21.96 | 40.33 ± 18.72 | 37.75 ± 19.48 [a] | 12.58 ± 9.95 [b] |
| SM5 | 62.75 ± 9.5 [b] | 49.92 ± 8.5 [b] | 55.08 ± 18.99 [b] | 37.33 ± 23.37 | 32 ± 16.28 | 34.33 ± 18.46 [a] | 11.17 ± 9.34 [b] |
| SM10 | 73.17 ± 7.28 [b] | 57.25 ± 13.34 [b] | 69.5 ± 21.6 [b] | 29.92 ± 29.96 | 26.58 ± 13.27 | 18.58 ± 12.06 [b] | 3.13 ± 5.33 [c] |
| | | | | Biscuits with the addition of cloves (*Eugenia caryophyllata*) | | | |
| SC | 17.57 ± 12.66 [a] | 72.14 ± 16.89 [a] | 29.14 ± 13.86 [a] | 37 ± 17.57 | 35.43 ± 14.52 [a] | 39.57 ± 22.26 [a] | 7.86 ± 11.07 [a,b] |
| SH1 | 56.42 ± 6.6 [b] | 43.08 ± 9.89 [b] | 55.75 ± 16.11 [b] | 49.33 ± 16.52 | 42.25 ± 16.71 [a] | 49.25 ± 14.7 [a] | 20.42 ± 17.08 [a] |
| SH3 | 65.92 ± 10.89 [b,c] | 44.5 ± 11.07 [b] | 54.58 ± 15.16 [b] | 38.75 ± 22.62 | 35.25 ± 17.39 [a] | 38.17 ± 13.89 [a,b] | 10.42 ± 11.2 [a,b] |
| SH5 | 70.83 ± 9.67 [c,d] | 50.9 ± 12.75 [b] | 56.75 ± 19.92 [b] | 36 ± 25.73 | 28.42 ± 12.62 [a,b] | 28.17 ± 9.93 [b,c] | 4.96 ± 6.72 [a,b] |
| SH10 | 77.83 ± 12.93 [d] | 55.75 ± 12.84 [a,b] | 55.75 ± 12.84 [b] | 31.17 ± 35.91 | 15.75 ± 12.56 [b] | 15.50 ± 10.97 [c] | 4.04 ± 6.99 [b] |
| | | | | Biscuits with the addition of cinnamon (*Cinnamomum verum*) | | | |
| SC | 17.57 ± 12.66 [a] | 72.14 ± 16.89 [a] | 29.14 ± 13.86 [a] | 37 ± 17.57 [a] | 35.43 ± 14.52 [a] | 39.57 ± 22.26 [a] | 7.86 ± 11.07 [a] |
| SS1 | 40.14 ± 18.84 [c] | 55.57 ± 15.53 [b] | 48.71 ± 8.22 [b] | 60.14 ± 19.76 [b] | 56.14 ± 15.2 [c] | 58.86 ± 15.98 [b] | 26.57 ± 19.27 [c] |
| SS3 | 68.86 ± 10.96 [b,c] | 47.00 ± 9.85 [b] | 48 ± 9.54 [b] | 65.57 ± 20.65 [b] | 52.43 ± 8.02 [b,c] | 63.43 ± 15.98 [b] | 30.29 ± 19.29 [c] |
| SS5 | 62.00 ± 14.28 [b,c] | 58.00 ± 15.31 [b] | 50.14 ± 13.95 [b] | 61.29 ± 24.03 [b] | 48.57 ± 11.77 [b,c] | 51.86 ± 23.13 [b] | 27 ± 18.65 [c] |
| SS10 | 76.71 ± 10.58 [b,c] | 47.57 ± 14.68 [b] | 54.43 ± 16.35 [b] | 38.86 ± 16.53 [b] | 34.86 ± 11.23 [b] | 39.43 ± 15.97 [a] | 12.29 ± 12.04 [b] |

Different letters indicate statistically significant ($p < 0.05$) differences; SC, control biscuit; SHo, biscuit with added grape seed flour (0, 3, 5, or 10%); SM, biscuit with added mint (0, 3, 5, or 10%); SH, biscuit with added clove (0, 3, 5, or 10%); and SS, biscuit with added cinnamon (0, 3, 5, or 10%).

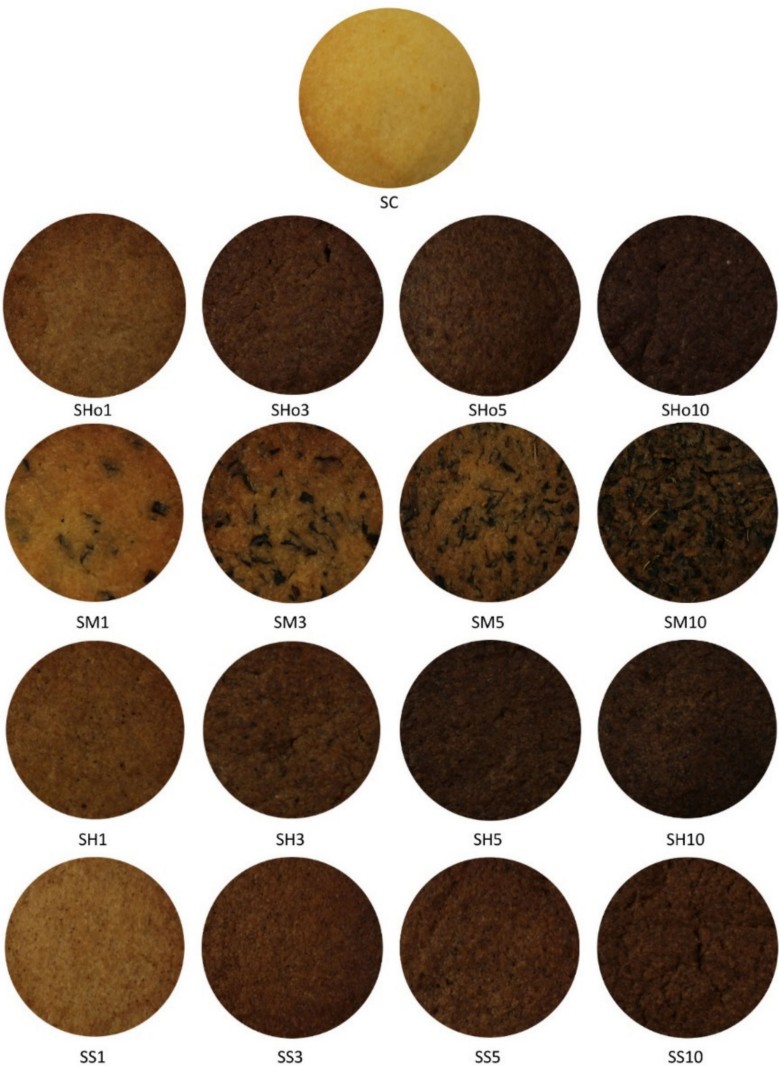

**Figure 1.** Colour changes of biscuits depending on the addition of spices/herbs. SC, control biscuit; SHo, biscuit with added grape seed flour (0, 3, 5, or 10%); SM, biscuit with added mint (0, 3, 5, or 10%); SH, biscuit with added clove (0, 3, 5, or 10%); and SS, biscuit with added cinnamon (0, 3, 5, or 10%). $p < 0.05$ $p < 0.05$ $p < 0.05$.

### 3.4. Colour

Sensory evaluation is very important in the acceptance of the food product, though it cannot be described as a fully objective method as it can be influenced significantly by external factors, including physic-psychical conditions of the panellists [40,41]. Sensory evaluation of colour in products was evaluated in terms of colour intensity. For better evaluation, the samples were evaluated instrumentally as well. The CIELAB system is commonly used for colour evaluation [42]. However, other methods of colour evaluation can be used in the food industry [14,43,44].

For colour analysis, we selected the RGB and HSV colour space and the whole biscuits were scanned, not just the defined area that is used in the CIELAB system. The simple intensity of the RGB channels does not provide a sufficiently accurate colour image to match human perception. Therefore, other recalculated parameters were used for comparison, which express colour in a way closer to human perception. Twelve colour features describing RGB and HSV colour space were used (features are in detail in chapter 2.6) for the comparison of products. The result is shown in Figure 2 using PCA analysis, which is a more suitable method for displaying a number of variables. Most of the variation in the data set can be explained by the first two principal components. Eigenvalue higher than 1.0

was for F1 and F2 (9.16 and 1.67). The first two components explain 76.3% and 90.23% of the cumulative variance. The relation of all measured colour features to factors are shown in Figure 3. The two-dimensional plot of loadings of Principal Component 1 and Principal Component 2 shows that colour features (Mean Red, Blue, Green, and Saturation) as well as colour intensity (Mean Intensity and Brightness) are not able to classify biscuits with spices/herbs addition from one another, as these parameters are closely linked. The HSV parameters (Hue, Saturation, Brightness) are better suited to biscuit classification, as these parameters are separate from one another (Figure 3).

The obtained results (Figure 2) clearly confirm the different colour of the control group in contrast to the fortified products. The different colouration was also confirmed for the separate group of products with the addition of mint. The sample with the addition of cinnamon in a concentration of 1.0% also had a similar colour. Products with grape flour and cloves form an interconnected group. Therefore, their colours do not differ significantly. Figure 2 also shows the distribution of groups in SM in connection with the increasing concentration of the addition.

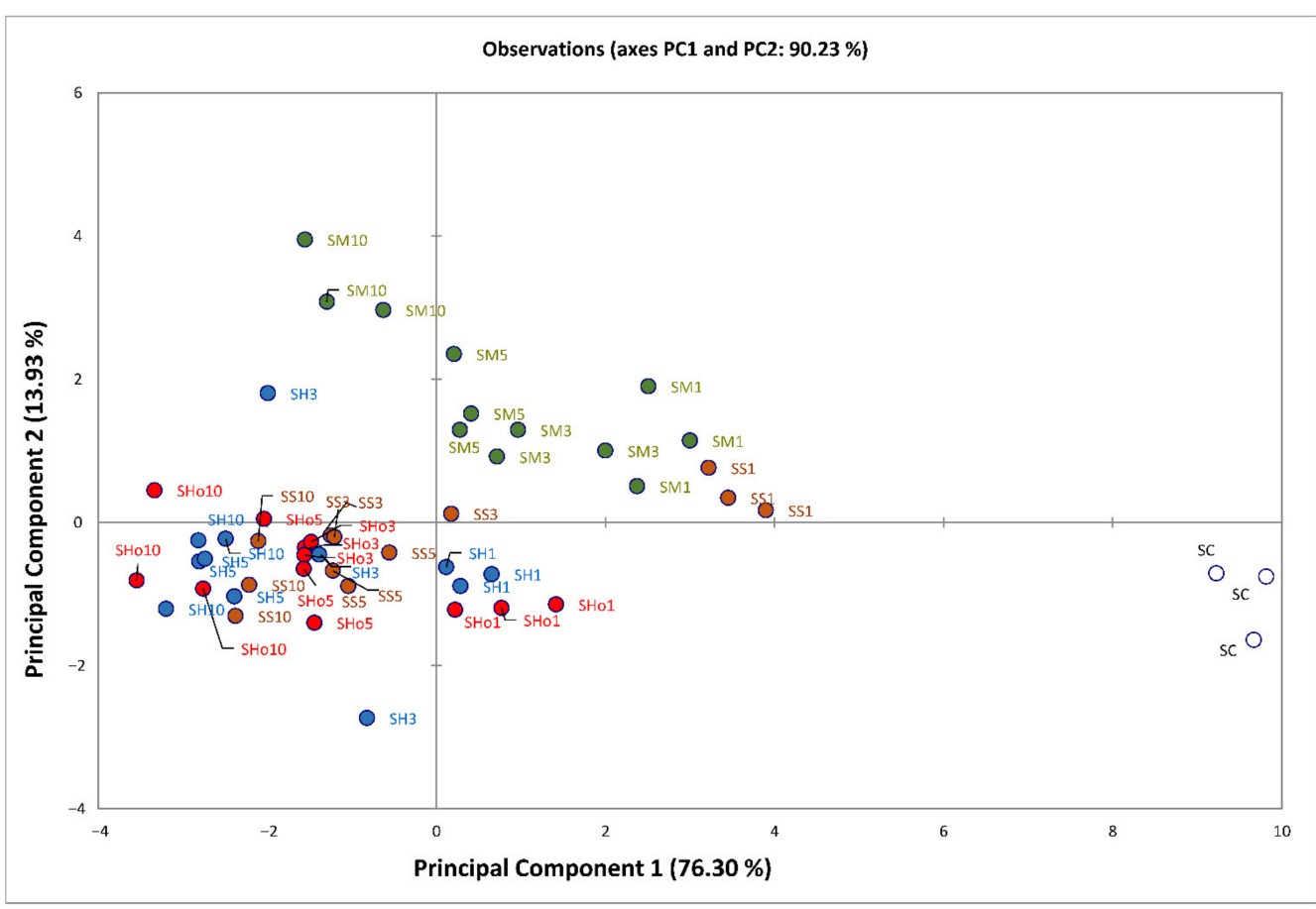

**Figure 2.** Colour evaluation of biscuits using PCA. SC, control biscuit; SHo, biscuit with added grape seed flour (0, 3, 5, or 10%); SM, biscuit with added mint (0, 3, 5, or 10%); SH, biscuit with added clove (0, 3, 5, or 10%); and SS, biscuit with added cinnamon (0, 3, 5, or 10%).

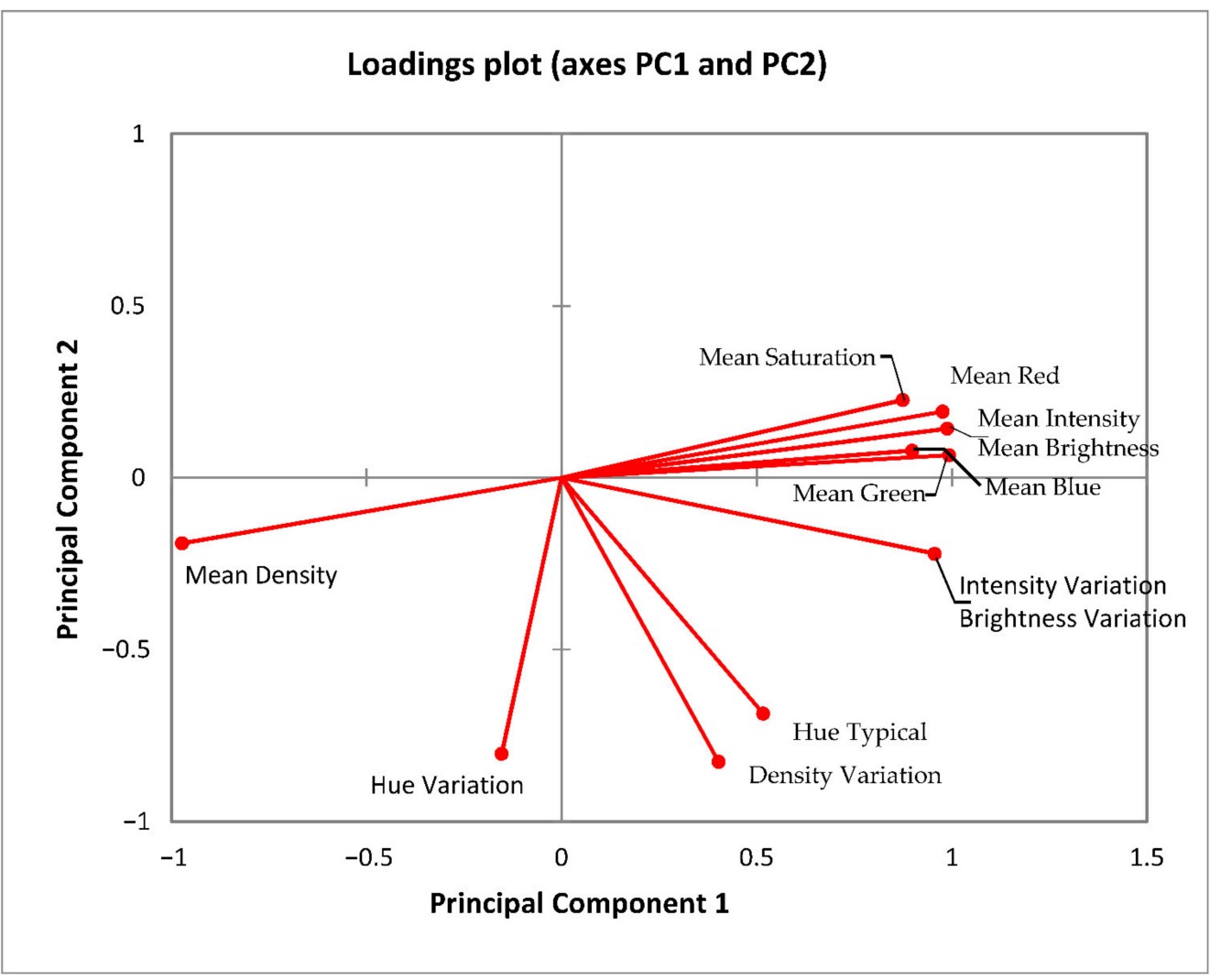

**Figure 3.** Similarities and dissimilarities of colour features for biscuits classification. The two-dimensional loadings plot of Principal Component 1 and Principal Component 2.

The resulting colour of the biscuits is the result of two ongoing processes, one of which is the Maillard reaction, in which a reaction between sugars and amino acids occurs. If the amount of protein is reduced, the Maillard reaction will also be reduced. This reaction takes place throughout the biscuit dough, but is more intense on the surface, which is promoted by high temperatures and low moisture content [45]. The colour changes due to fortification of the biscuits were also confirmed (Table 3, Figures 1 and 3). The darker colour of biscuits after fortification was also confirmed for cinnamon, mint, and cloves [34,37]. The colour arises from natural pigments such as anthocyanins and tocopherols [46] from the plant species, but it also confirmed effect of the addition of grape seeds to the colour of biscuits [47].

## 4. Conclusions

The results of the experiment confirm that the appropriate use of spices and herbs can increase the content of polyphenols and thus increase the antioxidant capacity in biscuits. The polyphenols were transferred to the product by spice, resulting in a significant increase in TPC in biscuits. This was reflected in an increase in antioxidant capacity. Samples with cloves showed the highest increase in antioxidant capacity, which also corresponds to the highest content of polyphenols in this spice. However, an increasing trend was observed for all samples tested. The highest values were measured in samples with the



10.0% addition of spices, but at this level of addition, products were not well accepted by consumers. The addition of spices and herbs into the biscuits also changes other indicators that are important for the overall evaluation of the products. It is, therefore, necessary to find the optimal balance between improving antioxidant properties and acceptability for consumers when creating recipes. From the point of view of the content of natural polyphenols in biscuits, their antioxidant capacity, influence on organoleptic properties, and overall acceptability for the customer, the use of cloves in a concentration of 3.0% proves to be the most suitable. The antioxidant capacity expressed by both DPPH (90.7%) and FRAP (102.23 µmol Trolox/g), is also higher than in other samples, even with higher additions. According to sensory evaluation, the product remains attractive to consumers. Based on the performed measurements, it can be stated that a reasonable addition of natural substances containing natural antioxidants improve the overall quality of biscuits.

**Supplementary Materials:** The following are available online at https://www.mdpi.com/article/10.3390/pr9081455/s1, Protocol S1: Total Polyphenols Content.

**Author Contributions:** The authors' responsibilities were as follows: Conceptualization, M.P.; methodology, M.P.; validation, Z.J., D.D. and O.K.; formal analysis, K.F.; investigation, K.F.; resources, B.T.; data curation, D.D.; writing—original draft preparation, O.K.; writing—review and editing, M.P., S.J. and Z.J.; visualization, O.K.; supervision, M.P. All authors have read and agreed to the published version of the manuscript.

**Funding:** This research received no external funding.

**Institutional Review Board Statement:** Not applicable.

**Informed Consent Statement:** Not applicable.

**Data Availability Statement:** Not applicable.

**Conflicts of Interest:** The authors declare no conflict of interest.

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
