# Peer review of "Biscuits Polyphenol Content Fortification through Herbs and Grape Seed Flour Addition"

_processes, doi:10.3390/pr9081455_

Round 1
Reviewer 1 Report
Dear authors,
thank for your research. The topic of this manuscript is intersting, and in accordance with the circular economy, too. However, the quality of the presentation is not high. I advice to once more work on the text. Please include all my suggestions in the corrected manuscript.
Abstract: there is no numerical findings. The part description of the obtained results is too short, mainly it is a description and the results are not compared with other authors.
The introduction is long, however it doesn't contain good explanation for used spices/herbs.
Line 32 One of them is the oxidation of fats, which affects other parameters. It causes oxidative stress which results in the development of rancidity, unpleasant taste and odour. - Combine and connect these two sentences together and mention the degradation products that give the taste of rancidity unpleasant taste and odour.
Line 50 – 76 The introduction should state the specific facts as to why these particular plants were selected. This part of the introduction is more for discussion because it describes too much what other authors got and their results.
Also, at the end of the introduction there is no written aim of the research.
Try to work also on paragraph titles in Materials and methods. In the section of materials and methods, there is no subchapter which contains used chemicals used and their manufacturers.
Why was wheat flour taken and not wholemeal? Why were these concentrations chosen? Explain.
Line 113 10 ml write 10 mL - redo throughout the paper
Line 114 (Whatman 1) write more clearly (grade 1)
After 10 minutes shaking, the sample was filtered (Whatman 1). – write clearly the sentence and mention equipment used for shaking.
Line 115 Na2CO3 – should be written Na2CO3 - check all chemical formulas in paper and rewrite
Line 115 75.0 g/l – should be 75.0 g/L – rewrite
Line 127 FeCl3_6H2O - rewrite the chemical formula and write it correctly
Line 132 20 ml of ethanol - what percentage – state
Line 133 0.1 mm ethanolic solution of DPPH – should be 0.1 mM
Line 137 and 138 the formula should be numbered
Line 153 The mentioned ISO norm must be listed in the references
In "Sensory analysis" any norms and references should be added.
Line 155 – 156 define a scale in more detail
Why at least some chemical quality parameters such as acid number, iodine number,% moisture were not measured? Higher percentage of moisture, ie the availability of water can affect the organoleptic properties of biscuits because spices/herbs contain microorganisms that can change these properties and the increased amount of available water allows microorganisms to grow.
2.7. Statistical Analysis - this subsection does not state how the results are presented and how the measurements were performed
Line 172 - Kruscal‐Walis test – should be Kruskal -Wallis test
The tables in the Results chapter are not well explained nor is stated what the figures shown represent.
(p<0.05) is not uniformly written in the paper - revise
Table 3 is not announced/mentioned in the text. It should also go under the subsection of sensory results.
The conclusion should be shorter and more concise.
The literature in References is not written according to the rules and the sequence is not respected.
Future directions: Since you mentioned shelf life and oxidation of biscuits, have you tried to do measurements of quality parameters during storage and antioxidant capacity because storage also leads to phenol degradation? I think you would get interesting results.
Reviewer 2 Report
- Paper is poorly written, insufficient details – major revision, nearly rejected
- Line 21 – standard – better to use word ‘control’
For Line 41, 42 and 43 – I will suggest this recent paper https://doi.org/10.3390/antiox10050721
- Line 61, 62 – change – The extract was also compared with the BHA which clearly demonstrated the antioxidant effect of mint.
- Line 61 to 76 explained the results – add in results and discussion part
- Line 115 – subscript
- Line 118 – add gallic acid lowest-highest concentration used in this experiment e.g. 0-200
- Part 2.2 and 2.3– add detailed description of every protocol (one by one) performed in this experiment in supplementary material e.g. concentration of used chemicals – insufficient details
- Line 173 to 187 – shows the seriousness of the authors - very unfortunate
- Provide all three values of TPC – full calculations – excel sheet
- Table 1, 2, 3 – adopt same pattern – required corrections
- Discussion – Authors forgot to add discussion part – this poorly is poorly written, they didn’t add the enough discussion which can attract or will be suitable for other researchers.
- Material and Methods part needs detailed description
Round 2
Reviewer 1 Report
Dear authors,
Thank you for the corrected version of the paper. The revised manuscript is much better than the first version, but some little things still need to be improved.
Introduction:
One of them is the oxidation of fats, which affects other parameters. It causes oxidative stress which results in the development of rancidity, unpleasant taste and odour. - Combine and connect these two sentences together and mention the degradation products that give the taste of rancidity unpleasant taste and odour (that was the first note) - the authors disobeyed and rewrote according to the given suggestion. Omit the word oxidative stress because it is excess and mention the degradation products that give the taste of rancidity.
Line 66,67,68 - Latin names must be written in italic (e.g. Mentha piperita L.)
The authors still did not work on paragraph titles in Materials and Methods.
Also didn't add subchapter which contains used chemicals used and their manufacturers. The chemicals used must be clearly indicated in the paper and not in the supplementary materials.
DPPH scavenging activity [%] = [(AbsDPPH ‐ Abssample)/AbsDPPH] – the equation should be written like in attached document.
Results and Disscusion
p < 0.05 – p should be written in italic. Redo throughout the paper (p < 0.05)
Line 288-291 Revise the sentence. It is not well written.
The literature in References is still not written according to the rules and the sequence is not respected. Rework!!!
ISO norm should be written like following example:
International Organization for Standardization (ISO). Sensory Analysis-Methodology-Method of Investigating Sensitivity of Taste; ISO 3972:2011; ISO: Geneva, Switzerland, 2011.

Reviewer 2 Report
This paper is suitable for publication. Moreover, I will suggest improve it before final submission.
Author Response
Dear reviewer,
Thanks for your valuable comments. We have also incorporate more suggestions from other reviewer and editor. We provided also some formal and gramatical correction.
Kind regards
Authors